**Data Availability Statement:** The data underlying this study is available from: https://www.researchgate.net/publication/344687962_Joint_

# Intra- and inter-rater reliability of joint range of motion tests using tape measure, digital inclinometer and inertial motion capturing

**Laura Fraeulin**[1]*, **Fabian Holzgreve**[1], **Mark Brinkbäumer**[2], **Anna Dziuba**[2], **David Friebe**[2], **Stefanie Klemz**[2], **Marco Schmitt**[2], **Anna-Lena Theis A.**[2], **Sarah Tenberg**[2], **Anke van Mark**[1], **Christian Maurer-Grubinger**[1], **Daniela Ohlendorf**[1]

**1** Institute for Occupational Medicine, Social Medicine and Environment Medicine, Goethe-University Frankfurt, Frankfurt am Main, Germany, **2** Institute of Sports Sciences, Goethe-University Frankfurt, Frankfurt am Main, Germany

* fraeulin@med.uni-frankfurt.de

## Abstract

### Background

In clinical practice range of motion (RoM) is usually assessed with low-cost devices such as a tape measure (TM) or a digital inclinometer (DI). However, the intra- and inter-rater reliability of typical RoM tests differ, which impairs the evaluation of therapy progress. More objective and reliable kinematic data can be obtained with the inertial motion capture system (IMC) by Xsens. The aim of this study was to obtain the intra- and inter-rater reliability of the TM, DI and IMC methods in five RoM tests: modified Thomas test (DI), shoulder test modified after Janda (DI), retroflexion of the trunk modified after Janda (DI), lateral inclination (TM) and fingertip-to-floor test (TM).

### Methods

Two raters executed the RoM tests (TM or DI) in a randomized order on 22 healthy individuals while, simultaneously, the IMC data (Xsens MVN) was collected. After 15 warm-up repetitions, each rater recorded five measurements.

### Findings

Intra-rater reliabilities were (almost) perfect for tests in all three devices (ICCs 0.886–0.996). Inter-rater reliability was substantial to (almost) perfect in the DI (ICCs 0.71–0.87) and the IMC methods (ICCs 0.61–0.993) and (almost) perfect in the TM methods (ICCs 0.923–0.961). The measurement error (ME) for the tests measured in degree (˚) was 0.9–3.3˚ for the DI methods and 0.5–1.2˚ for the IMC approaches. In the tests measured in centimeters the ME was 0.5–1.3cm for the TM methods and 0.6–2.7cm for the IMC methods. Pearson correlations between the results of the DI or the TM respectively with the IMC results were significant in all tests except for the shoulder test on the right body side (r = 0.41–0.81).

**Funding:** The author(s) received no specific funding for this work.

**Competing interests:** The authors have declared that no competing interests exist.

## Interpretation

Measurement repetitions of either one or multiple trained raters can be considered reliable in all three devices.

## Introduction

Range of motion (RoM) measurements are often used to assess functional mobility [1, 2]. However, unassisted assessments of RoM, which are still performed in medical assessments, are subjective and, therefore, lack reliability [3, 4]. Nevertheless, standardized and objective RoM tests can be useful tools in choosing and evaluating therapeutic treatments in patients with issues of the musculoskeletal system [5, 6]. However, good intra- and inter-rater reliabilities should also be ensured for the respective test procedures using measurement devices in order to properly evaluate the therapy progress [2]. Both the intra- and inter-rater reliability depend on the experience of the rater, the health status of the participants, the accuracy of the instrument, its exact use and the respective test protocol [2, 7].

In clinical practice goniometers, digital inclinometers (DI) and tape measures (TM) are in frequent use [1, 2, 5, 6]. These pieces of equipment are easy to handle and are also cost and time effective. Recently, the smartphone has also received considerable attention as a measuring device and several applications have been compared to goniometer data, (reviewed by Keogh et al. [8]). The authors showed mostly adequate intra- and inter-rater reliability as well as validity for smartphone devices on all joints assessed. However, the intra-rater reliability of the DI has been shown to be equivalent [9] to the goniometer, while the inter-rater reliability was found to be superior [10, 11], since the exact placement of goniometer can be difficult [7, 12, 13]. The reliabilities of the DI and TM procedures, however, do differ [2, 14]; for example, in evaluating the flexion of the lumbar spine reliability was found to be inconsistent for several (slightly) modified versions of the Schoeber method using a TM [15–17] and an inclinometer placed on the spine [18, 19].

A relatively new and frequently applied method is inertial motion capturing (IMC); this allows for precise and objective kinematic measurement in real world environments [3, 20, 21]. So far, IMC has been used by researchers when assessing functional RoM [22, 23] but, as yet, not for assessing the maximal range of motion which is held in static positions. Inertial sensors, like they are used in the MVN system by Xsens, combine signals of accelerometers, gyroscopes and magnetometers to determine the position, acceleration and orientation of body segments in space [24]. When multiple sensors are applied, joint angles can be calculated for all three body dimensions by means of biomechanical models [25].

In motion capturing, the optical approach (OMC) is considered the gold standard of kinematic measurements [26, 27]. However, as OMC is executed in the laboratory with several high precision cameras, it is scarcely available and is also expensive. Although research of the validity of IMC systems relies on small sample sizes [28], concurrent findings suggest that the IMC systems deliver good to excellent data especially in the frontal and sagittal planes [24, 27, 29, 30] and in slower and less complex movements [20, 31]. For example, comparing the Xsens system to the Optotrak system, Robert-Lachaine et al. [31] have shown that in the Xsens system the mean root mean square error (RMSE) on all joints was 1.2° in short functional movements compared to 2.8° in faster and more complex tasks (p≤0.001). They also showed, that the differences between the Xsens and Optotrak systems were significantly more attributed to discrepancies in the biomechanical model, rather than technological issues in estimating the

orientation of segments (RMSE <5˚ in manual handling tasks); this has been supported by the findings of Zhang et al. [28]. The validation studies for the Xsens system have already been published in 2016 [32] and 2013 [28] and since then the calibration procedure and data processing protocols have been further developed.

In the present study, we chose to evaluate five traditional joint RoM tests. In these tests, RoM data are captured in static poses, mostly in the sagittal plane (fingertip-to-floor test, Thomas test, retroflexion of the trunk) and in the frontal plane (lateral inclination) while only for the shoulder test is a combination of planes used (Fig 1).

The aim of this study was to collect joint range of motion data using low cost device methods (DI and TM) and more objective kinematic data using the Xsens system. For both approaches the intra- and inter-rater reliability are calculated in order to determine their practicability for medical assessment.

## Methods

### Study protocol

The aim of this study was to compare data obtained with either the DI [5, 7, 13] or TM [33, 34] with the data derived from the IMC system in order to assess the reliability of five RoM tests. Thus, we chose the following five tests (Fig 1) which depict the flexibility of the trunk in all three dimensions as well as the shoulder and the hip: the shoulder test modified after Janda [35], the modified Thomas test [5], retroflexion of the trunk after Janda in a modified version [35], the fingertip-to-floor test [33] and lateral inclination [12].

The study was approved by the ethics committee of Goethe University (2018–46) Frankfurt am Main, Germany.

### Subjects

22 healthy subjects (12f/10m) with an average age of 25 years ± 2 volunteered to participate in this prospective study. The subjects were on average 174.1 cm ± 9.8 tall and weighed 66.6 kg ± 11.3. The average body mass index (BMI) was 21.9 ± 2.0 and all subjects were right-handed. All participants provided written informed consent. Two raters (sports scientists, B.A. and B.Sc., respectively) were intensively trained in the use of the measuring systems. They were responsible for performing the range of motion tests and positioning the DI and the TM.

To take part in the study, subjects had to be aged between 18 and 30 years. Exclusion criteria consisted of relevant operations to, or surgical stiffening, of the musculoskeletal system, relevant artificial joint replacement, severe diseases such as ankylosing spondylitis, chronic destructive joint diseases, multiple sclerosis, myodystrophic or neurodegenerative diseases, congenital malpositions of the musculoskeletal system or an acute heriated disc. In addition, the intake of muscle relaxants or other drugs that influence the elasticity of the musculature and pregnancy were considered as contra indicators.

### Materials

**Inertial Motion Capture (IMC).** The MVN BIOMECH Link system from Xsens (Enschede, Netherlands) was used for kinematic data collection. This system provides position, orientation and angle information, in addition to acceleration and velocity parameters, of the entire human body by means of 17 motion sensors, each consisting of acceleration and inertial sensors as well as gyroscopes. The sampling rate of the system is 240 Hz and the measurement error is specified by the manufacturer as ± 1%. The corresponding software displays the collected data in real-time using kinematic motion reconstruction as a biomechanical

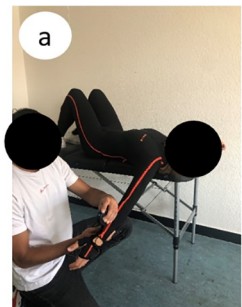
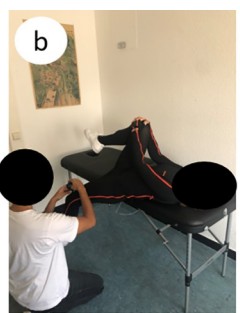
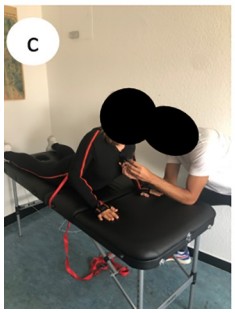
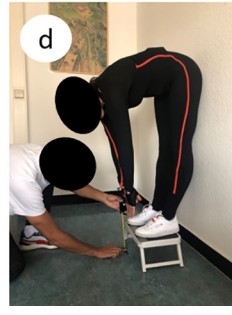
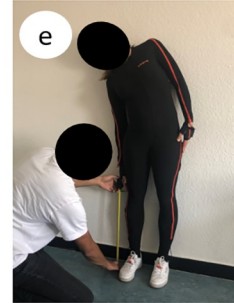

**Fig 1. The range of motion measurements.** a) Shoulder test modified after Janda on the left-hand body side, b) modified Thomas test on the left-hand body side, c) retroflexion of the trunk after Janda in the modified version, d) fingertip-to-floor test and e) lateral inclination on the right-hand body side. The measurements were simultaneously recorded in the Xsens system (subjects wears the measurement suit) and the low cost devices.

model. Due to the arrangement of the sensors within a full body suit (available in various sizes), the system is comfortable to wear without restricting the freedom of movement. In the study, when the position to be measured was reached, a marker was set in the recording session. The recording was executed in the multi-level scenario, which is recommended by Xsens when measurements are not uniquely performed on a flat floor (for our measurements subjects partly lay on a bench or stood on a step, thus, the multi-level scenario was the best setting applicable for these tests). Where subjects stood on even ground, the single-level scenario was chosen. After the measurements were taken, the "HD reprocessing" filter was applied to all recordings, this is provided in the MVN Analyze software and offers the best possible data quality according to the manufacturer.

**Digital Inclinometer (DI).**    A digital inclinometer (Model: AcumarTM DIGITAL INCLI-NOMETER Model ACU002 / Lafayette Instrument Company / Lafayette / USA) was used for the angle measurement in the modified Thomas test, in the retroflexion of the trunk modified after Janda and also in the shoulder test modified after Janda. As the DI displays only integers, the absolute measurement error is 0.3˚ [36].

**Tape Measure (TM).**    A commercially available tape measure was used to measure the distance between the finger and the ground and for lateral flexion. The measuring tape had a double measuring scale (inch and cm) with the smallest increments being 0.1 cm. The hard surface and the associated reduced buckling behavior enabled exact measurements to be made. The distances were recorded to the nearest 0.1 cm.

## Measurement protocol

Prior to the measurements, subjects dressed in the measurement suits of the IMC system which were then calibrated. The five RoM tests described were carried out by the test persons

in a randomized order. Since the modified Thomas test, the lateral inclination and the shoulder test modified after Janda were tested on both body sides, this study included a total of eight tests. For each test, 25 repetitions were performed which were recorded simultaneously by the IMC system and the DI or IMC. The first 20 repetitions were recorded by the first rater but not included in any calculations as they were included as warm-up in order to control for acute effects [37, 38]. For rater one, measurement 21–25 were included in the analysis [39]. Subsequently, the second rater measured another five repetitions. The order of the raters was chosen at random.

The raters were responsible for the correct positioning of the subjects at the beginning of each measurement and the application of either the DI or TM. When the subjects had reached the position to be measured, an additional investigator gave a verbal signal for the simultaneous measurement to be taken for the IMC and DI or TM. In the IMC, markers were set in the software which were checked after the recordings. The raters read out loud the angles or distances of the DI or the TM, respectively. The order of the raters was also randomized. The correct execution of the exercises was monitored by an additional investigator who underwent training at the same time as the raters.

## Range of motion measurements

In Fig 1 the performed RoM tests are presented. Further detailed information can be found in the methodology paper of Holzgreve et al. [13].

**Shoulder test modified after Janda.** In contrast to the test by Janda, in this study the elbow was stretched (Fig 1A) and the raters placed the inclinometer proximal to the processus styloideus radii on the radius. While the test person was lying on a treatment couch, the shoulder joint was free and the arm was lowered in a controlled manner until the tension of the musculature terminated the movement. For each measurement, the rater moved the subject's arm and decided when the defined position had been reached. The test was performed separately for each arm. The joint centers of the wrist and the humerus head were extracted from the IMC. The arm length (distance between the humerus head and wrist) and the height difference between the humerus head and the wrist joint were calculated. The angle was calculated by the sin-1 from height/length. The recording was executed in the multi-level scenario and subsequently HD reprocessed.

**Modified Thomas test.** During the measurement session of the modified Thomas test (Fig 1B), the 0˚ alignment of the pelvis was checked by the raters after every fifth measurement. The inclinometer was then placed on the thigh, proximal to the patella, to determine the joint angle [40, 41]. The leg remained in the same position during all measurements and while changing the raters; the test was also performed separately for each leg. For the analysis of the IMC data, the flexion/extension angle of the hip joint was used. The recording was executed using the multi-level scenario and subsequently reprocessed (not HD).

**Retroflexion of the trunk modified after Janda.** The retroflexion of the trunk test, which was modified after Janda (Fig 1C), was determined by placing the inclinometer on the proximal part of the sternum. At the instructor's command, the test person adopted the position to be measured, maintained it for a few seconds and the angle was measured by the rater. For comparison with the IMC system, the orientation angle (y-axis) of the sternum was analyzed. The recording was executed using the multi-level scenario and subsequently HD reprocessed.

**Fingertip-to-floor test.** For the fingertip-to-floor tests (Fig 1D), subjects adopted the standardized position on a 15 cm high stool; this ensured that flexible persons could also execute the test with full RoM. For the execution, it was checked that the knees were always stretched and that the index fingers of both hands were brought together. The distances between the floor and

the fingertip were measured using a conventional measuring tape. For the corresponding IMC data, the distance between the hand segments and the foot segments were calculated; to do so, the data of the left and right side were averaged for hand and foot segments.

**Lateral inclination.** The lateral inclination was executed in a standardized standing position. Sagittal fluctuations in the lateral inclination were eliminated by leaning the back against a wall. The ipsilateral hand of the body side to be measured had to be guided distally, directly along the body. An investigator confirmed that the test person kept their knees straight, that their heels were not lifted off the ground and that their backs were always leant against the wall. If the position to be measured was actively reached by the test person, the rater measured the lateral fingertip-to-floor distance using a measuring tape. Each body side was measured separately. For comparison with the IMC system, the distance between the hand segment and the floor was calculated. The recording was executed in the single level scenario and afterwards HD reprocessed.

## Statistical analysis

Statistical analysis was performed using Matlab R2020a, Microsoft Excel 2016, IBM SPSS Statistics 25 and BIAS (Version 11) software programs. The relevant joint angles were provided in the Xsens software except for the specific angle used in the shoulder test. For this test, Matlab was used to calculate the distances between the segments and the angles. In addition, all necessary IMC data were analyzed in Matlab and exported as Excel-files similar to the DI and the TM data.

For the intra-rater reliability of rater 1 and rater 2, the last five measurements of each test were taken into account. Inter-rater reliability was analyzed by comparing the mean values of the last five measurements of rater 1 and rater 2. Reliabilities were calculated by means of intraclass correlation coefficients (ICC) using the BIAS software.

Intra-rater reliability (ICC) was assessed according to Bland and Altmann [42]:

$$\mathrm{ICC} = (\mathrm{m\ SSB} - \mathrm{SST})/(\mathrm{m}-1)\ \mathrm{SST},$$

where m is the number of observations per subject, SST the total sum of squares, and SSB the sum of squares between persons. Measurements are considered reliable if differences between the two measurements of the same person are small compared to the differences between the individuals.

The measurement error (ME) is the square root of the average of the subject-specific variances:

$$\mathrm{ME} = \sqrt{\left[\sum_i \sum_j (\overline{Y} - Y_{ij})^2 / (m-1)\right]/i}$$

Repeatability was calculated as the mean difference between two measurements of the same subject, which can be estimated as $\sqrt{2} \cdot \mathrm{ME}$ [43].

The coefficient of repeatability (CoR) was calculated as an estimation of the minimum detectable change: $1.96 \times \sqrt{2} \cdot \mathrm{ME}$.

For inter-rater reliability for the low cost device methods, the "two-way mixed, single-measure" ICC(3,1) according to Shrout and Fleiss [44] was used, where ICC(3,1) values were estimated via this random effects model:

$$Y_{ij} = \mu + s_i + \alpha_j + e_{ij}, \quad i = 1, 2, \ldots, 22, j = 1, 2$$

with $Y_{ij}$ the j-th observation of the i-th person, $\mu$ the fixed effect, $s_i$ the random effect of the i-th person, iid, $N(0, \sigma_s^2)$, $\alpha_j$ the fixed rater effect of the j-th rater, $e_{ij}$ the measurement error, iid, $N(0, \sigma_e^2)$.

Measurements are considered reliable if differences between the observers are small compared to the differences between the individuals. The intra-class correlation coefficient ICC (3,1) measures the relation of the variance that is attributed to a random factor as

$ICC(3,1) = [\sigma^2_s - \sigma^2_r]/[\sigma^2_s + \sigma^2_r + \sigma^2_e]$, with $\sigma^2_r$ being the variance of the rater.

Both intra- and inter-rater reliability coefficients (ICCs) were classified by means of the method suggested by Landis and Koch [45]: ICCs 0–0.20 = "slight", 0.21–0.40 = "fair", 0.41–0.60 = "moderate", 0.61–0.80 = "substantial", 0.81–1.00 = "(almost) perfect".

As the vast majority of data was normally distributed, the Pearson correlation coefficient was used to show the relationship between the measurement systems using BIAS. Only data of which the synchronicity could be assured was included. The correlation coefficients were classified according to Evans [46]: r <0.2 = "poor", 0.2–0.4 = "weak", 0.4–0.6 = "moderate", 0.6–0.8 = "strong" and >0.8 = "optimal" correlations.

## Results

Intra-rater reliability for all tests using the DI or the TM, respectively, showed (almost) perfect results (Table 1). The same was demonstrated for the IMC measurement system (Table 2). For the RoM tests measured in centimeters, the ME ranged from 0.5–1.3cm in the TM method and 0.6–2.7cm in the IMC method. The repeatability ranged from 0.7–1.9cm in the TM method and from 0.9–3.9cm in the IMC method. In the tests measured in degrees (˚) the ME was 0.9–3.3˚ in the DI method and 0.5–1.2˚ in the IMC method. The repeatability was 1.2–4.7˚ in the DI method and 0.5–2.8˚ in the Xsens system.

Inter-rater reliability (Table 3) for the DI and the TM protocols revealed substantial to (almost) perfect agreement. This can also be described for the IMC testing.

The relationship between the two methods is illustrated in Fig 2 for each test. The correlations between the low cost device methods and the IMC method were shown to be moderate or better and except for the shoulder test on the right-hand body side statistically significant

**Table 1. Intra-rater reliability for both raters using the DI and TM.**

| DI/TM | | Thomas test | | Shoulder test | | Retroflexion | Lateral Inclination | | Fingertip-to-floor |
|---|---|---|---|---|---|---|---|---|---|
| | Measuring system | DI | | DI | | DI | TM | | TM |
| | Body side | left | right | left | right | ------ | left | right | ------- |
| | Rater | | | | | | | | |
| ICC | Rater 1 | 0.976 | 0.987 | 0.942 | 0.945 | 0.914 | 0.887 | 0.975 | 0.965 |
| | Rater 2 | 0.962 | 0.955 | 0.925 | 0.947 | 0.886 | 0.974 | 0.962 | 0.951 |
| 95% Confidence Interval | Rater 1 | [0.95; 0.99] | [0.97; 0.99] | [0.90; 0.98] | [0.91; 0.98] | [0.86; 0.97] | [0.82; 0.96] | [0.96; 0.99] | [0.94; 0.98] |
| | Rater 2 | [0.93; 0.98] | [0.92; 0.98] | [0.88; 0.97] | [0.91; 0.98] | [0.82; 0.95] | [0.96; 0.99] | [0.94; 0.99] | [0.92; 0.98] |
| P-value | Rater 1 | 0.001 | 0.001 | 0.001 | 0.001 | >0.001 | >0.001 | >0.001 | >0.001 |
| | Rater 2 | 0.001 | 0.001 | 0.001 | 0.001 | >0.001 | >0.001 | >0.001 | >0.001 |
| ME | Rater 1 | 0.9˚ | 0.9˚ | 1.8˚ | 2.2˚ | 2.7˚ | 1.1 cm | 0.6 cm | 1.1 cm |
| | Rater 2 | 1.1˚ | 1.4˚ | 2˚ | 2.2˚ | 3.3˚ | 0.5 cm | 0.8 cm | 1.3 cm |
| Repeatability | Rater 1 | 1.6˚ | 2˚ | 2.6˚ | 3.1˚ | 3.8˚ | 1.6 cm | 0.9 cm | 1.5 cm |
| | Rater 2 | 1.3˚ | 1.2˚ | 2.9˚ | 3.1˚ | 4.7˚ | 0.7 cm | 1.1 cm | 1.9 cm |
| CoR | Rater 1 | 3.1˚ | 3.9˚ | 5.1˚ | 6.1˚ | 7.5˚ | 3.1 cm | 1.8 cm | 2.9 cm |
| | Rater 2 | 2.4˚ | 2.4˚ | 5.7˚ | 6.1˚ | 9.2˚ | 1.4 cm | 2.2 cm | 3.7 cm |

DI = digital inclinometer; TM = tape measure; IMC = inertial motion capture; ME = measurement error; CoR = coefficient of repeatability.

ICCs were rounded to three digits after the decimal point when ICCs lay between 0.899–1.00 to show that they were still below 1. The units for measurement error, repeatability and CoR are centimeters where a TM was used and degree˚ where a DI was used.

**Table 2. Intra-rater reliability for both raters using the IMC.**

| IMC | | Thomas test | | Shoulder test | | Retroflexion | Lateral Inclination | | Fingertip-to-floor |
|---|---|---|---|---|---|---|---|---|---|
| | Measuring system | DI | | DI | | DI | TM | | TM |
| | Body side | left | right | left | right | ------ | left | right | ------- |
| | Rater | | | | | | | | |
| ICC | Rater 1 | 0.996 | 0.990 | 0.921 | 0.970 | 0.972 | 0.985 | 0.924 | 0.958 |
| | Rater 2 | 0.994 | 0.982 | 0.899 | 0.938 | 0.974 | 0.984 | 0.942 | 0.921 |
| 95% Confidence Interval | Rater 1 | [0.99; 1.00] | [0.98; 1.00] | [0.87; 0.97] | [0.95; 0.99] | [0.95; 0.99] | [0.98; 0.99] | [0.88; 0.97] | [0.93; 0.98] |
| | Rater 2 | [0.99; 1.00] | [0.97; 0.99] | [0.83; 0.97] | [0.90; 0.98] | [0.96; 0.99] | [0.97; 0.99] | [0.91; 0.98] | [0.87; 0.97] |
| P-value | Rater 1 | >0.001 | >0.001 | >0.001 | >0.001 | >0.001 | >0.001 | >0.001 | >0.001 |
| | Rater 2 | >0.001 | >0.001 | >0.001 | >0.001 | >0.001 | >0.001 | >0.001 | >0.001 |
| ME | Rater 1 | 0.5° | 0.7° | 1.5° | 1.3° | 1.1° | 0.6 cm | 1.6 cm | 2.2 cm |
| | Rater 2 | 0.7° | 1.1° | 1.6° | 2° | 1.2° | 0.7 cm | 1.5 cm | 2.7 cm |
| Repeatability | Rater 1 | 0.7° | 1° | 2° | 1.9° | 1.5° | 0.9 cm | 2.2 cm | 3.1 cm |
| | Rater 2 | 1° | 1.6° | 2.3° | 2.8° | 1.6° | 1 cm | 2.2 cm | 3.9 cm |
| CoR | Rater 1 | 1.4° | 2° | 3.9° | 3.7° | 2.9° | 1.8 cm | 4.3 cm | 6.1 cm |
| | Rater 2 | 2° | 3.1° | 4.5° | 5.5° | 3.1° | 2 cm | 4.3 cm | 7.6 cm |

DI = digital inclinometer; TM = tape measure; IMC = inertial motion capture; ME = measurement error; CoR = coefficient of repeatability.

ICCs were rounded to three digits after the decimal point when the ICCs lay between 0.899–1.00 to show that they were still below 1. The units for measurement error, repeatability and CoR are centimeters where a TM was used and degree° where a DI was used.

(Table 4). While the fingertip-to-floor test exhibited an optimal correlation, the analysis of the lateral inclination and the modified shoulder test on the right-hand body side revealed strong correlations. The retroflexion of the trunk, the modified shoulder test of the right-hand body side and the Thomas test were found to correlate in a moderate manner.

The main findings of Tables 1–4, concerning the intra- and inter-rater reliabilities and the correlations, are summarized in Table 5.

## Discussion

The aim of this study was to provide both intra- and inter-rater reliabilities for five joint range of motion tests measured with low cost devices (TM or DI) and an IMC system. The results show, that intra-rater reliability was (almost) perfect in all tests for all three devices; whilst

**Table 3. Inter-rater reliability for using the TM and DI.**

| Inter-rater Reliability | | Thomas test | | Shoulder test | | Retroflexion | Lateral Inclination | | Fingertip-to-floor |
|---|---|---|---|---|---|---|---|---|---|
| | Measuring system | DI | | DI | | DI | TM | | TM |
| | Body side | left | right | left | right | ------ | left | right | ------- |
| | Rater | | | | | | | | |
| ICC | DI/TM | 0.87 | 0.72 | 0.80 | 0.71 | 0.84 | 0.961 | 0.938 | 0.923 |
| | IMC | 0.86 | 0.78 | 0.61 | 0.81 | 0.910 | 0.993 | 0.975 | 0.965 |
| 95% Confidence Interval | DI/TM | [0.77; 0.96] | [0.55. 0.85] | [0.57; 0.91] | [0.43; 0.88] | [0.66; 0.93] | [0.91; 0.98] | [0.86; 0.97] | [0.82; 0.97] |
| | IMC | [0.69; 0.94] | [0.55; 0.90] | [0.21; 0.83] | [0.58; 0.92] | [0.79; 0.96] | [0.98; 1.00] | [0.94; 0.99] | [0.92; 0.99] |
| P-value | DI/TM | 0.001 | 0.001 | 0.001 | 0.001 | >0.001 | >0.001 | >0.001 | >0.001 |
| | IMC | >0.001 | >0.001 | 0.003 | >0.001 | >0.001 | >0.001 | >0.001 | >0.001 |

DI = digital inclinometer; TM = tape measure; IMC = inertial motion capture; ME = measurement error.

ICCs were rounded to three digits after the decimal point when the ICCs lay between 0.899–1.00 to show that they were still below 1.

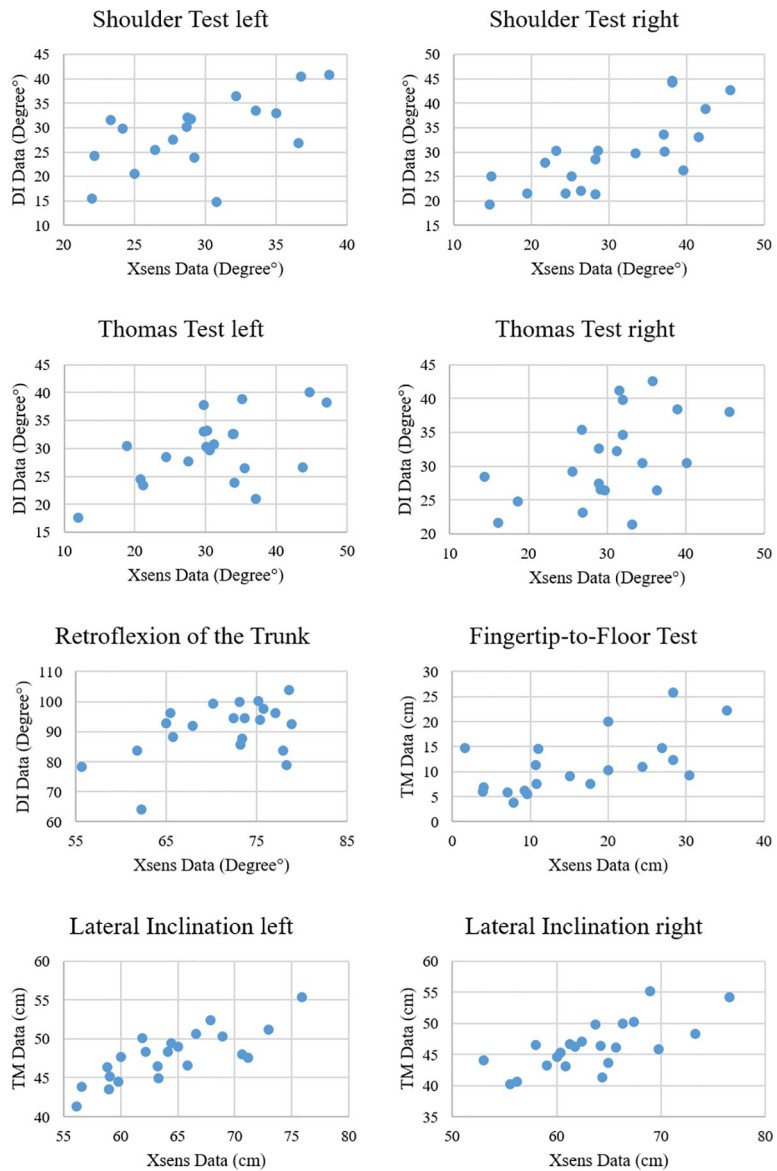

*DI = Digital Inclinometer; TM = Tape Measure

**Fig 2. The relationship between the low cost device methods and the IMC method.** Here the means of the last five measurements of rater 1 were used for the plot.

**Table 4. Pearson correlations between the DI–IMC or TM–IMC measurement systems.**

| Correlations | Thomas test | | Shoulder test | | Retroflexion | Lateral Inclination | | Fingertip-to-floor |
|---|---|---|---|---|---|---|---|---|
| Body side | left | right | left | right | ------ | left | right | ------- |
| Measurement Systems | DI—IMC | DI—IMC | DI—IMC | DI—IMC | DI—IMC | TM—IMC | TM—IMC | TM—IMC |
| Correlation Coefficient (r) | 0.49* | 0.53* | 0.54* | 0.41 | 0.52* | 0.79* | 0.73* | 0.81* |
| n | 21 | 20 | 19 | 20 | 21 | 20 | 22 | 22 |

DI = digital inclinometer; TM = tape measure; IMC = inertial motion capture.

The 21st repetition of rater 1 was used exemplarily; only measurements where synchronicity could be ensured were included. Statistical significance is indicated by *.

**Table 5. Summary of the correlations, intra- and inter-rater reliabilities.**

| Summary | | Thomas test | | Shoulder test | | Retroflexion | | Lateral Inclination | | Fingertip-to-floor | |
|---|---|---|---|---|---|---|---|---|---|---|---|
| | Meas. system | DI | IMC | DI | IMC | DI | IMC | TM | IMC | TM | IMC |
| | Body side | | | | | | | | | | |
| **Correlation** | left | moderate | | moderate | | moderate | | strong | | optimal | |
| | right | | | | | | | | | | |
| **Intra-rater Reliability** | left | (almost) perfect | | | | | | | | | |
| | right | | | | | | | | | | |
| **Inter-rater Reliability** | left | (almost) perfect | substantial | substantial | | (almost) perfect | | (almost) perfect | | (almost) perfect | |
| | right | substantial | | (almost) perfect | | | | | | | |

DI = digital inclinometer; TM = tape measure; IMC = inertial motion capture.

inter-rater reliability was shown to be substantial to (almost) perfect in the DI and the IMC methods and (almost) perfect in the TM methods (Table 5). Consequently, the measurement repetitions of all three devices with either one or multiple trained raters can be considered reliable.

While the TM and the DI are in frequent use, the relatively new IMC systems have not, as yet, been used for static ROM assessment. This study compliments the field of IMC science, being the first to provide intra- and inter-rater reliabilities on static RoM measurements. Although the IMC approach does not show the exact same angles or distances as a TM or a DI (e.g. distance calculation between the hand and foot sensors or the distance from the fingertip to floor measured via TM), the correlations between the methods were moderate to strong (Table 4, Fig 2). It can therefore be assumed that the same or at least a very similar mobility construct was measured. Typically, IMC systems are used for the kinematic measurement of motions, for which reliabilities have been provided. van der Straaten et al. [24] used an Xsens device (Awinda, 60Hz) for the examination of joint angles of the trunk, pelvis, hip, knee and ankle in all three degrees of freedom in a single leg squat. For the sagittal hip RoM the results showed within-session, between-session and between operator reliabilities ICCs of 0.9, 0.86 and 0.86 respectively. Relative to the total RoM executed in the single leg squat, the proportional standard error of measurement (%SEM) in the hip joint was 18%– 20% and the minimal detectable change (MDC) was 8–11°. In the Thomas test of the current study, we also measured the hip RoM in the sagittal plane and demonstrated a considerably lower ME (0.5–1.1°) and CoR (1.4–3.1°). This is possibly due to the more precise Xsens system MVN Link which has a sampling rate of 240Hz and the static nature of the Thomas test which improves measurement accuracy [20].

The results of the TM and DI methods can be evaluated with regard to the concurrent evidence. Regarding the measurement of shoulder mobility, literature provides a large range of different RoM measurements [14, 47–50]; given the fact that the shoulder is a highly mobile joint, from a functional point of view, different degrees of freedom are of interest. In these studies, reproducibility has been shown to differ [14, 50, 51]; for example de Winter et al. [48] report ICCs ranging from 0.28–0.90 for the inter-rater reliability and a MDC of 20–25° for glenohumeral abduction and external rotation measured via DI. However, to our knowledge so far, no study has researched the precise RoM used in the current approach, which can be described as a combination of external rotation and abduction of the shoulder. Based on the presented results, this approach produces reliable results and an acceptable CoR ranging from 5.1–6.1° for the DI method and from 3.9–5.5° for the IMC method.

In the Thomas test the present findings of substantial to (almost) perfect reproducibility are supported by concurrent evidence [5, 11]. In addition, the CoRs of 2.4–3.9˚ in the DI method and 1.4–3.1˚ in the IMC method support a precise application of the test protocol in medical assessment. However, the Thomas test must, in general, be applied cautiously in medical assessments in obese people [52]; controlling for hip flexion [53] and testing proximal of the patella may be influenced by the curvature of the thigh and the subcutaneous fat tissue. Nevertheless, in the present study, this was not an issue as the subjects had a rather low BMI (21.9 ± 2.0).

In the retroflexion of the trunk, the available results on intra- and inter-rater reliabilities for measuring the retroflexion on different bony landmarks on the back are rather mixed [2, 54–57]. Mellin et. al. [52] have shown that the lying position is the most reliable method available (correlations between repetitions: *r* 7.2–9.2). However, they applied the inclinometer on the spine, whereas in this study the DI was placed on the sternum as it is easy to palpate and provides a solid base for the DI. Up to the present day, no other study has aimed to evaluate the extension of the spine via taking measurements from the sternum. Although reliability was (almost) perfect, the ME in the DI method ranged from 2.71–3.30˚ and the CoR was 7.5–9.2˚; these values can both be considered relatively high as RoM gains in the extension of the spine may only be small. Therefore, we recommend an even clearer definition of the bony landmarks when using the DI assessment. The IMC method, on the other hand, was shown to be a precise approach (ME (1.1–1.2˚) and CoR (2.9–3.1˚)).

In the lateral inclination the present results add to those from assessments of lateral spinal flexion in a study using a bubble inclinometer [58] and a measurement table against which the subject had to lean [59]; both studies produced (almost) perfect ICCs for the intra-rater reliability and inter-rater reliability, respectively. In the current study, the low ME and good repeatability values add to the practicability of the TM method in medical assessments. The ME of 0.5–1.1cm for the TM is even smaller than previously reported by Inger et al. [60] (2.6cm); here their SDC for this method was 7.3cm [60] which is considerably greater than the CoR of 1.8–2.2cm found in the present investigation. One explanation for this may be the thorough warm-up procedure used in the present investigation. In the IMC method, the CoR was 4.3cm; this could be due to the fact that the position data calculated by signal processing in IMC systems is not entirely accurate [61].

In the fingertip-to-floor test, current evidence shows mixed results. While three studies [1, 34, 62] confirm the current, very good intra-rater reliabilities (ICCs 0.97–0.99), two studies [4, 16] have reported a low reproducibility, possibly due to differences in the study design. Merrit et al. [4] recorded the measurements for the intra-rater reliability on three different days with only one instructional repetition; this, however, is not sufficient to control for acute effects. In addition, the subjects in this study were 18–65 years old. The lack of a warm-up might also explain the poor repeatability Gill et al. [16] described (coefficient of variation: 14.1), since the subjects maintained the flexed position while a rater took repeated measurements. Better reproducibility is presented by Ekedahl et al. [6] who report a 4.5cm MDC, which is similar to the present finding of 2.9–3.7cm CoR.

When selecting measuring instruments for medical assessment, the current findings, in general, support the easy to use low cost devices since the described MEs, repeatabilities and CoRs support to the good reproducibility results. It is only in the retroflexion of the trunk, that the application of the DI on the sternum might be too imprecise to capture actual changes in mobility. The IMC method also showed satisfying results in the tests measured in degrees, especially in the spinal movement where it could provide precise and reliable data. However, when measuring distances between body segments, it should be kept in mind, that position estimation with inertial sensors is not, as yet, entirely precise [61]. While the MVN Link system

by Xsens is expensive and probably too time consuming for use in clinical practice, future research should aim to evaluate cheaper and more practical IMC systems.

## Author Contributions

**Conceptualization:** Laura Fraeulin, Fabian Holzgreve, Daniela Ohlendorf.

**Data curation:** Laura Fraeulin, Fabian Holzgreve, Mark Brinkbäumer, Anna Dziuba, David Friebe, Stefanie Klemz, Marco Schmitt, Anna-Lena Theis A., Sarah Tenberg, Christian Maurer-Grubinger.

**Formal analysis:** Laura Fraeulin, Christian Maurer-Grubinger.

**Investigation:** Laura Fraeulin, Fabian Holzgreve, Mark Brinkbäumer, Anna Dziuba, David Friebe, Stefanie Klemz, Marco Schmitt, Anna-Lena Theis A., Sarah Tenberg.

**Methodology:** Laura Fraeulin, Fabian Holzgreve, Mark Brinkbäumer, Anna Dziuba, David Friebe, Stefanie Klemz, Marco Schmitt, Anna-Lena Theis A., Sarah Tenberg, Daniela Ohlendorf.

**Resources:** Daniela Ohlendorf.

**Software:** Laura Fraeulin, Daniela Ohlendorf.

**Supervision:** Anke van Mark, Daniela Ohlendorf.

**Writing – original draft:** Laura Fraeulin.

**Writing – review & editing:** Fabian Holzgreve, Mark Brinkbäumer, Anna Dziuba, David Friebe, Stefanie Klemz, Marco Schmitt, Anna-Lena Theis A., Sarah Tenberg, Anke van Mark, Christian Maurer-Grubinger, Daniela Ohlendorf.

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
