## [Decision Letter · Decision Letter 0]

3 Aug 2020

PONE-D-20-15948

Intra- and Interrater Reliability of Range of Motion Tests: Tape Measure and Digital Inclinometer compared to Inertial Motion Capture

PLOS ONE

Dear Dr. Maltry,

Thank you for submitting your manuscript to PLOS ONE. After careful consideration, we feel that it has merit but does not fully meet PLOS ONE’s publication criteria as it currently stands. Therefore, we invite you to submit a revised version of the manuscript that addresses the points raised during the review process.

ACADEMIC EDITOR: Please see the reviewer comments for detailed feedback on your manuscript. Please consider all mentioned aspekts while revising your manuscript. Your study is of high interested but major changes are required (rationale; data analysis) before possible consideration of publication.

We look forward to receiving your revised manuscript.

Kind regards,

Juliane Müller, PhD

Academic Editor

PLOS ONE

Journal Requirements:

2.We note that you have indicated that data from this study are available upon request. PLOS only allows data to be available upon request if there are legal or ethical restrictions on sharing data publicly. For information on unacceptable data access restrictions, please see http://journals.plos.org/plosone/s/data-availability#loc-unacceptable-data-access-restrictions.

3. Your ethics statement must appear in the Methods section of your manuscript. If your ethics statement is written in any section besides the Methods, please move it to the Methods section and delete it from any other section. Please also ensure that your ethics statement is included in your manuscript, as the ethics section of your online submission will not be published alongside your manuscript.

Reviewers' comments:

Reviewer's Responses to Questions

**Comments to the Author**

1. Is the manuscript technically sound, and do the data support the conclusions?

Reviewer #1: Partly

Reviewer #2: Partly

2. Has the statistical analysis been performed appropriately and rigorously? 

Reviewer #1: Yes

Reviewer #2: No

3. Have the authors made all data underlying the findings in their manuscript fully available?

Reviewer #1: No

Reviewer #2: Yes

4. Is the manuscript presented in an intelligible fashion and written in standard English?

Reviewer #1: Yes

Reviewer #2: Yes

5. Review Comments to the Author

Reviewer #1: General comments

This study seeks to examine an interesting question regarding the reliability of several range of motion tests that have traditionally been used in clinical physiotherapy practice to more recent advances in wearable IMU technology. Such a question therefore could be of interest to developers of these technologies as well as to researchers and practitioners in the field of physiotherapy and exercise science. The paper is quite well-written and presented throughout and details some quite comprehensive results. However, I have a number of major reservations for the authors to consider.

Specific comments

Title: I’m not sure the title completely encompasses the data provided in the manuscript. In that you have compared the Xsens inertial motion capture data to that of the tape measure and digital inclinometer data, this is some type of relationship, perhaps even partially a validity study. I therefore suggest you add in the phrase “Relationship” somewhere in the title as well as in the aims of the study, especially as the first two lines of results you have reported in your abstract (line 42 and 43), describe these relationships.

Line 46 – 47: I’m not sure if this interpretation is the best for your paper as that would mean that one of these tools is considered a criterion method of the assessment of joint range of motion. As traditional 3D motion capture systems such as Vicon are typically considered the criterion method for assessing joint range of motion, I would prefer you focus here on the reliability of each system rather than the relationship between them. Specifically, if the reliability of the Xsens is greater than that of the more traditional methods, that might be one reason to support its use in clinical practice.

Line 74: you may wish to refer to this systematic review which has summarised the validity of a class of inertial device (smart devices) to criterion measures as well as the intratester and intertester reliability of traditional and inertial methods of joint range of motion assessment. https://pubmed.ncbi.nlm.nih.gov/31067247/ This paper may provide some better context to the study and provide some data with which to compare your results within your discussion section.

Line 85 – 86: I was surprised you only included the digital inclinometer and tape measure here and not the goniometer as one of the traditional measurement tools. You mentioned goniometers on line 70 and I am therefore highly surprised they were not included in this study, especially with some of the recent research that has compared goniometers to the inertial sensors found in smart phones, such as that summarised in the systematic review I highlighted in the previous comment.

Line 117 – 130: while a comparison of traditional 3D motion capture methods to the inertial motion capture system is of theoretical and practical interest, this was not performed in the study and no data was provided in the introduction about how well these measurements are related in the literature. Therefore, I am not sure of the practical utility of this comparison. As you are not comparing any of these systems to what is considered the criterion method of traditional 3D motion capture, you can’t specifically state one system is more valid or accurate than the other. Further, even if the inertial motion capture system is more reliable than the traditional methods, the cost of the system, time required to collect data and the complicated Matlab code needed to generate the data means that such an approach is not clinically feasible. Further, such an approach even seems more complicated than traditional 3D motion capture in which the software provides the joint range of motion data relatively easily after the joint markers positions have been tracked during the movement.

Line 176 – 177: I may have missed it, but what is meant by HD reprocessed?

Line 219 – 239: it was great to see some measures of absolute and not just relative reliability included in this study. However, can you provide a clearer definition of measurement error and repeatability and perhaps some references to other studies that have used these measures compared to other absolute measurement error scores such as mean difference and limits of agreement, coefficient of variation, root mean square error etc?

Line 243: what is meant by “optimal effect size” here and in Table 5? I have seen that you have defined this term earlier in the statistics section, but I can’t see anywhere in the tables any effect size data. Further, if you’re comparing the difference in the scores between the different devices in the study, I would have thought that the smaller effect size, the greater the similarity/relationship between the two devices. This would mean in such a study you would like to see very small effect size differences rather than large effect size differences, meaning that an optimal effect size would be very small rather than large.

Table 2 – 4: I would suggest that the p-value row is placed after the 95% confidence interval row in these three tables. Further, I would suggest that for any of your measurement errors or repeatability values that are measured in degrees, two decimal places is way beyond the precision of measurement. I would therefore suggest that such data have one decimal place maximum. This would also apply to any distances that are measured in millimetres.

Table 4: I was wondering why there was no measurement error/repeatability data provided in this table.

Line 278: you cannot state that any of your devices or methods are more accurate than one another as you did not compare them to any criterion method such as traditional 3D motion capture. As I’ve stated before, you can only say that such devices were more similar or had stronger relationships to each other.

Lines 334 – 336: what are the units of measurement for this measurement error?

Line 336 – 338: would this small potential sagittal plane arm swing be much of a difference in the results? If so, was there any testing method or data analysis method you could have used to minimise the potential effect of the arm sway?

Line 342 – 345: could potential difference in the participants or the level of training of assessors perhaps be influential here in these between study differences?

Overall Discussion: it might be useful refer back to this systematic review https://pubmed.ncbi.nlm.nih.gov/31067247/ to compare your findings to that of the reliability and to a slightly lesser extent the validity research for measuring joint range of motion. When you are talking about your absolute reliability scores such as measurement error, it would also be useful to refer back to the wider literature to what is considered the minimum clinically important difference for these joint range of motion and whether the measurement error is smaller or larger than the MCID.

Overall references: you have included many up-to-date and relevant references for the IMU literature, but it appears many of the references for the inclinometer and tape measure methods are dated. Is this because little research has been conducted on these methods over the last two decades as the universal goniometer is the most common form of joint range of motion assessment in clinical practice?

Reviewer #2: The presented study investigates the inter- and intra-rater reliability of three different techniques to assess RoM based on five standardized movement tests. The authors highlight that validity and reliability is key for adequate practical application of RoM assessments in a clinical environment. They therefore compare the results of two low cost techniques (digital inclinometer and tape measure) against a kinematic measurement system (inertial motion capture). Based on the results (ICC and measurement errors), the authors recommend both low cost systems for the use in medical assessments.

Major issues:

In its current form it is not clear enough stated what is the precise aim of the investigation. Is the IMC system assumed to be the “gold standard” to validate the use of the two low cost techniques? If so, the validity and reliability of the IMC system against the laboratory “gold standard” for movement assessment (until today, that is still a marker based optoelectrical 3D-camera system) needs to be stated and discussed much more thoroughly.

The experimental design is in general thoroughly described; however, some aspects will benefit of more details to fully understand all performed comparisons (see points below).

In terms of conducted statistical tests it is recommended to also incorporate Bland-Altman analyses (with corresponding plots). This would allow to assess systematic and random errors between the systems given in its actual measurement units. A more critical discussion in regards to the findings of this study is required.

Minor issues/comments:

- Line 75: if the IMC is your gold standard, then you will need to provide further details of its validity/reliability compared to highly standardized laboratory methods.

- Line 92: this figure only shows the graphical representation of the IMC software. It would be helpful to also provide pictures of an exemplary participant where the actual measurements (IMC vs DI or IMC vs TM) are performed

- Line 101: its recommended to always use the same amount of decimal points (And in this case even rounding up to the full year would be fine for this kind of data)

- Line 117 and following: here the reader needs more information about the system. Are there other resources about measurement precision than given by the manufacturer (independent scientific investigations). The provided statement is too general. Measurement precision depends on various calculations and will much likely depend on the site and the movement (as discussed by the authors in part later in the discussion)

- Line 128: What is meant by HD reprocessed? This needs to be introduced/explained

- Line 137: Is there a source for that?

- Line 152: This is not clear for the reader. This means 15 familiarization trials before 5 actual measurements trials? But later your spearman calculations are based on up to 550 number of repetitions.

- Line 168 and following: as mentioned above, an exemplary picture with the methods applied would much likely be appreciated by the readers. Same applies for the other tests.

- Line 174: what is meant by “length in 3D”?

- Line 223: How much did the data deviate from normal distribution. This also affects the outcome/validity of the ICC calculations.

- Line 257 and following: The tables are sometimes difficult to read. It is recommended to separate the different statistical tests by thicker lines (or in a different way), e.g. between “Measurement error” and “95% Confidence interval”;

- All tables: used abbreviations need to be explained in a legend for each table

- Table 1: How are these high ”n’s” possible? That is not sufficiently explained before

- Table 2: The unit of measurement for “Measurement Error” and “repeatability” is missing, or? Both outcomes are not well described within the manuscript (and only short in the given reference)

- Table 4: the denotation “DI/TM” might be changed to “DI or TM”

- Line 276: language: why is a comparison justified? This is not well expressed. Low Spearman’s rho results (especially for the Thomas test) need to be further discussed.

- Line 279 and following: but there are more potential methodological reasons for that.

- Line 281 and following: Statement not clear

- Line 331: So in other words the comparison is potentially based on invalid data! That should be investigated and clearly stated

- Line 334 and following: measurement units?

- Line 356: If these methods are adequate to measure individual training progress, then it would be good to discuss how precise exactly they are. What is the minimal detectable change? Statement regarding the identification of individual training progress might however be beyond the possibilities of this study.

6. PLOS authors have the option to publish the peer review history of their article (what does this mean?). If published, this will include your full peer review and any attached files.

Reviewer #1: **Yes: **Justin Keogh

Reviewer #2: No

---

## [Author Response · Author response to Decision Letter 0]

16 Oct 2020

Response to Editor:

We have followed the instructions for formatting in the revision, uploaded the data in Researchgate and provided an DOI and included the Ethics statement only in the Methods Section.

Response to Reviewer 1:

Thank you very much for taking your time to assess our work and your suggestion. Please find the response to your comments below. In general, we learned, that there has been a misunderstanding of the aim of the study. We did not intend to conduct a validity study, which we apparently failed to describe more precisely in the first version of this manuscript, possibly because we are non-native English speakers. In the revised version, we emphasized on rephrasing the entire manuscript more distinct. We aimed at conducting range of motion data of low cost devices and IMC in order to obtain intra- and interrater reliabilities for both methods. 

Reviewer #1: General comments

This study seeks to examine an interesting question regarding the reliability of several range of motion tests that have traditionally been used in clinical physiotherapy practice to more recent advances in wearable IMU technology. Such a question therefore could be of interest to developers of these technologies as well as to researchers and practitioners in the field of physiotherapy and exercise science. The paper is quite well-written and presented throughout and details some quite comprehensive results. However, I have a number of major reservations for the authors to consider.

Specific comments

Title: I’m not sure the title completely encompasses the data provided in the manuscript. In that you have compared the Xsens inertial motion capture data to that of the tape measure and digital inclinometer data, this is some type of relationship, perhaps even partially a validity study. I therefore suggest you add in the phrase “Relationship” somewhere in the title as well as in the aims of the study, especially as the first two lines of results you have reported in your abstract (line 42 and 43), describe these relationships.

Thank you for the suggestion. We rephrased the title: “Intra- and Interrater Reliability of Joint Range of Motion Tests using Tape Measure, Digital Inclinometer and Inertial Motion Capturing”

Line 46 – 47: I’m not sure if this interpretation is the best for your paper as that would mean that one of these tools is considered a criterion method of the assessment of joint range of motion. As traditional 3D motion capture systems such as Vicon are typically considered the criterion method for assessing joint range of motion, I would prefer you focus here on the reliability of each system rather than the relationship between them. Specifically, if the reliability of the Xsens is greater than that of the more traditional methods, that might be one reason to support its use in clinical practice.

Thank you for the suggestion, we considered that when we rewrote the entire abstract.

Line 74: you may wish to refer to this systematic review which has summarised the validity of a class of inertial device (smart devices) to criterion measures as well as the intratester and intertester reliability of traditional and inertial methods of joint range of motion assessment. https://pubmed.ncbi.nlm.nih.gov/31067247/ This paper may provide some better context to the study and provide some data with which to compare your results within your discussion section.

Thank you for the suggestion, we have included the study in the introduction.

Line 85 – 86: I was surprised you only included the digital inclinometer and tape measure here and not the goniometer as one of the traditional measurement tools. You mentioned goniometers on line 70 and I am therefore highly surprised they were not included in this study, especially with some of the recent research that has compared goniometers to the inertial sensors found in smart phones, such as that summarised in the systematic review I highlighted in the previous comment.

Previously to our study we had long discussions in our team on how best to evaluate the chosen range of motion tests. We are well aware that the goniometer is used a lot in ROM measurements. However, it was difficult in the included tests to apply the goniometer properly, since as you also stated in your systematic review, the typical goniometer is not long enough or the adjustments to bony landmarks is difficult. For example, in the retroflexion of the trunk it would be hard to choose bony landmarks and to properly apply the goniometer, which also accounts for the shoulder test. Most importantly, we searched the literature for measurement accuracy of the goniometer and the DI. Our findings showed, that the DI is at least as accurate as the goniometer or even higher in some studies. While the intrarater reliability seem to be similar, the interrater reliability has been shown to be lower. We have included this in the introduction section now.

Line 117 – 130: while a comparison of traditional 3D motion capture methods to the inertial motion capture system is of theoretical and practical interest, this was not performed in the study and no data was provided in the introduction about how well these measurements are related in the literature. Therefore, I am not sure of the practical utility of this comparison. As you are not comparing any of these systems to what is considered the criterion method of traditional 3D motion capture, you can’t specifically state one system is more valid or accurate than the other. Further, even if the inertial motion capture system is more reliable than the traditional methods, the cost of the system, time required to collect data and the complicated Matlab code needed to generate the data means that such an approach is not clinically feasible. Further, such an approach even seems more complicated than traditional 3D motion capture in which the software provides the joint range of motion data relatively easily after the joint markers positions have been tracked during the movement.

Thank you for referring to the optical motion capture systems, we realized, we should have explained that more clearly in the first place. We tried to expand on this topic in the revised version of the manuscript. Also, the Xsens software provides real time joint angles and position data as well. So the Thomas Test and the retroflexion of the trunk at least could also be evaluated without the matlab code. However, the specific angle of the shoulder and the distances needed to be calculated in matlab, but this would also be the case in the Vicon system. 

Line 176 – 177: I may have missed it, but what is meant by HD reprocessed?

Thanks for mentioning. We clarified this in the manuscript: “Afterwards, on all recordings the “HD reprocessing” filter was applied, which is provided in the MVN Analyze software and offers the best possible data quality according to the manufacturer.”

Line 219 – 239: it was great to see some measures of absolute and not just relative reliability included in this study. However, can you provide a clearer definition of measurement error and repeatability and perhaps some references to other studies that have used these measures compared to other absolute measurement error scores such as mean difference and limits of agreement, coefficient of variation, root mean square error etc?

Thank you for the suggestion, we rewrote the section of statistical analyses much more detailed. As we have described before, we did not calculate validity eg. with Bland Altmann Plots an LoA. This is beacuse the low cost devices and the IMC system do not measure exactly the same construct. For example, in the tape measure tests we used the distances from the hand sensor, which is placed centrally on the back of the hand. In the lateral flexion we could calculate the distance to the floor because subjects were standing on even ground and we could use the single level scenario. In the finger-tip-to floor test we had to use the multi level scenario because subjects were standing on a little bench and not on even ground. In this scenario, a calculation to the floor is not possible. Therefore, we calculated the distance between the hand sensor and the foot sensor. It is also impossible to gain data of the fingertips in the IMC. Our aim was to best reproduce the construct used in the TM device, so the aimed range of motion is reflected likewise. Nevertheless, in the revised version of this manuscript we have included scatterplots which show the relationship between the methods (Fig2).

Line 243: what is meant by “optimal effect size” here and in Table 5? I have seen that you have defined this term earlier in the statistics section, but I can’t see anywhere in the tables any effect size data. Further, if you’re comparing the difference in the scores between the different devices in the study, I would have thought that the smaller effect size, the greater the similarity/relationship between the two devices. This would mean in such a study you would like to see very small effect size differences rather than large effect size differences, meaning that an optimal effect size would be very small rather than large.

Thanks for mentioning, the term effect size was wrongly used. We tried to clarify this in the entire manuscript. We are talking about Pearson correlations when we try to show, that the DI or TM measure similar constructs as the IMC. We aim at high correlations because this means that the DI or TM data shows a high similarity to those we calculated in Matlab using the IMC data. We did not calculate any differences. 

Table 2 – 4: I would suggest that the p-value row is placed after the 95% confidence interval row in these three tables. Further, I would suggest that for any of your measurement errors or repeatability values that are measured in degrees, two decimal places is way beyond the precision of measurement. I would therefore suggest that such data have one decimal place maximum. This would also apply to any distances that are measured in millimetres.

P-value has been moved in the line after the CI.

Measuremt Errors and Repeatability has been changed to max one decimal place.

There are no distances in this manuscript measured in millimeters.

Table 4: I was wondering why there was no measurement error/repeatability data provided in this table.

We have discussed this in detail with our statistics department and the colleague assured us that no measurement error is calculated for intra-rater reliability.

Line 278: you cannot state that any of your devices or methods are more accurate than one another as you did not compare them to any criterion method such as traditional 3D motion capture. As I’ve stated before, you can only say that such devices were more similar or had stronger relationships to each other.

Thank you for mentioning, we rewrote the discussion section avoiding classifications as “better” or “worse”.

Lines 334 – 336: what are the units of measurement for this measurement error?

The units are cm for the distances and ° for angular data. We have included this in the tables.

Line 336 – 338: would this small potential sagittal plane arm swing be much of a difference in the results? If so, was there any testing method or data analysis method you could have used to minimise the potential effect of the arm sway?

Line 342 – 345: could potential difference in the participants or the level of training of assessors perhaps be influential here in these between study differences?

Thank you for this question, we have now expanded on this: “While three studies [1, 44, 45] confirm the current, very good intra-rater reliabilities (ICCs 0.97 - 0.99), two studies [4, 13] have reported a low reproducibility, possibly due to differences in the study design. Merrit et al. [4] recorded the measurements for the intra-rater reliability on three different days but with only one instructional repetition, which is not enough to control for acute effects. Also, the subjects in this study were 18-65 years old. The lack of warm-up might also explain the poor repeatability Gill et al. [13] described (coefficient of variation: 14.1), since subjects maintained the flexed position while a rater took repeated measurements.”

Overall Discussion: it might be useful refer back to this systematic review https://pubmed.ncbi.nlm.nih.gov/31067247/ to compare your findings to that of the reliability and to a slightly lesser extent the validity research for measuring joint range of motion. When you are talking about your absolute reliability scores such as measurement error, it would also be useful to refer back to the wider literature to what is considered the minimum clinically important difference for these joint range of motion and whether the measurement error is smaller or larger than the MCID.

As described above we now focused more on the reliability instead of the validity.Thank you for mentioning the MCID. We initially considered referring to such values, but a deeper literature research showed that there are numerous MCID values even sometimes for the same joint ROM. Copay et al. summarized this in their 2Part Review doi: 10.2106/JBJS.RVW.17.00160. But we added comparisons to MDC values in the fingertip to floor test and the lateral inclination. 

Overall references: you have included many up-to-date and relevant references for the IMU literature, but it appears many of the references for the inclinometer and tape measure methods are dated. Is this because little research has been conducted on these methods over the last two decades as the universal goniometer is the most common form of joint range of motion assessment in clinical practice?

Of course, the IMU literature has been published more recently, since it has been developed only a few years ago and its accuracy has reached scientific standard only recently. This is also true for the smartphone applications. For the DI method, standard reliability research which we focused on initially has been conducted much earlier, since this is obviously a more traditional method and the tests are basic RoM tests. For example in the Thomas test, the DI method is still frequently used in research on clinical applications; we added 2 more recent references:

1. Kim WD, Shin D. Correlations Between Hip Extension Range of Motion, Hip Extension Asymmetry, and Compensatory Lumbar Movement in Patients with Nonspecific Chronic Low Back Pain. Med Sci Monit. 2020;26:e925080.

2. Roach SM, San Juan JG, Suprak DN, Lyda M, Bies AJ, Boydston CR. Passive hip range of motion is reduced in active subjects with chronic low back pain compared to controls. Int J Sports Phys Ther. 2015;10(1):13-20.

Also 1 publication on the intrarater reliablity of a DI and a goniometer was included; showing that both are excellent.

Roach S, San Juan JG, Suprak DN, Lyda M. Concurrent validity of digital inclinometer and universal goniometer in assessing passive hip mobility in healthy subjects. Int J Sports Phys Ther. 2013;8(5):680-8.

Response to Reviewer 2:

Thank you very much for taking your time to assess our work and your suggestion. Please find the response to your comments below. In general, we learned, that there has been a misunderstanding of the aim of the study. We did not intend to conduct a validity study, which we apparently failed to describe more precisely in the first version of this manuscript, possibly because we are non-native English speakers. In the revised version, we emphasized on rephrasing the entire manuscript more distinct. We aimed at conducting range of motion data of low cost devices and IMC in order to obtain intra- and interrater reliabilities for both methods in order to give advice for clinical applications.

Reviewer #2: The presented study investigates the inter- and intra-rater reliability of three different techniques to assess RoM based on five standardized movement tests. The authors highlight that validity and reliability is key for adequate practical application of RoM assessments in a clinical environment. They therefore compare the results of two low cost techniques (digital inclinometer and tape measure) against a kinematic measurement system (inertial motion capture). Based on the results (ICC and measurement errors), the authors recommend both low cost systems for the use in medical assessments.

Major issues:

In its current form it is not clear enough stated what is the precise aim of the investigation. Is the IMC system assumed to be the “gold standard” to validate the use of the two low cost techniques? If so, the validity and reliability of the IMC system against the laboratory “gold standard” for movement assessment (until today, that is still a marker based optoelectrical 3D-camera system) needs to be stated and discussed much more thoroughly.

The experimental design is in general thoroughly described; however, some aspects will benefit of more details to fully understand all performed comparisons (see points below).

In terms of conducted statistical tests it is recommended to also incorporate Bland-Altman analyses (with corresponding plots). This would allow to assess systematic and random errors between the systems given in its actual measurement units. A more critical discussion in regards to the findings of this study is required.

We did not include Bland Altmann plots, because we are aware that the low cost devices and the IMC system does not measure exactly the same construct. For example, in the tape measure tests we used the distances from the hand sensor, which is placed centrally on the back of the hand. In the lateral flexion we could calculate the distance to the floor because subjects were standing on even ground and we could use the single level scenario. In the finger-tip-to floor test we had to use the multi level scenario because subjects were standing on a little bench and not on even ground. In this scenario, a calculation to the floor is not possible. Therefore, we calculated the distance between the hand sensor and the foot sensor. It is also impossible to gain data of the fingertips in the IMC. Our aim was to best reproduce the construct used in the TM device, so the aimed range of motion is reflected likewise. Nevertheless, in the revised version of this manuscript we have included scatterplots which show the relationship between the methods (Fig2).

Minor issues/comments:

- Line 75: if the IMC is your gold standard, then you will need to provide further details of its validity/reliability compared to highly standardized laboratory methods.

Thank you for the suggestion. Although this is not a validity study we now included information on the comparison of the OMC and the IMC in the introduction section. 

Line 92: this figure only shows the graphical representation of the IMC software. It would be helpful to also provide pictures of an exemplary participant where the actual measurements (IMC vs DI or IMC vs TM) are performed

Thank you for mentioning. We now included a revised version of Fig1, where the exact measurements on an exemplary subject are shown. The subject wears the measurement suit and a rater measures angles and distances.

-Line 101: its recommended to always use the same amount of decimal points (And in this case even rounding up to the full year would be fine for this kind of data)

Thanks for the hint. Done.

Line 117 and following: here the reader needs more information about the system. Are there other resources about measurement precision than given by the manufacturer (independent scientific investigations). The provided statement is too general. Measurement precision depends on various calculations and will much likely depend on the site and the movement (as discussed by the authors in part later in the discussion).

Thank you for mentioning, we have expanded on this in the introduction section.

Line 128: What is meant by HD reprocessed? This needs to be introduced/explained

We have added explanations. 

Line 137: Is there a source for that?

Included in the method section.

Line 152: This is not clear for the reader. This means 15 familiarization trials before 5 actual measurements trials? But later your spearman calculations are based on up to 550 number of repetitions.

We have tried to clarify that section. “In each test 25 repetitions were performed which were recorded simultaneously by the IMC system and the DI or IMC. The first 20 repetitions were recorded by the first rater but not included in any calculations as they were included as warm up in order to control for acute effects [32, 33]. For rater one, measurement 21 – 25 were included in the analysis [34]. Subsequently, the second rater measured another five repetitions. The order of the raters was randomly chosen.”

Thanks for also mentioning the correlations. There was a mistake we know corrected. The new correlations were executed on only one repetition per subject. See the description of the new Table 4: “The 21st repetition of rater 1 was used exemplarily; only measurements where synchronicity could be ensured were included.”

Line 168 and following: as mentioned above, an exemplary picture with the methods applied would much likely be appreciated by the readers. Same applies for the other tests.

See new Fig 1.

Line 174: what is meant by “length in 3D”?

Now rephrased: “The arm length (distance between humerus head and wrist)…

Line 223: How much did the data deviate from normal distribution. This also affects the outcome/validity of the ICC calculations.

In the first version of the manuscript there was a mistake in the calculations of the correlations. Now we used only one repetition of each subject to exemplarily show the correlation. Here, all data was normally distributed. In the ICC calculations five measurements of each rater were included.

Line 257 and following: The tables are sometimes difficult to read. It is recommended to separate the different statistical tests by thicker lines (or in a different way), e.g. between “Measurement error” and “95% Confidence interval”;

Thanks for mentioning, we reformatted the tables.

All tables: used abbreviations need to be explained in a legend for each table

Done.

Table 1: How are these high ”n’s” possible? That is not sufficiently explained 

Thank you for mentioning. This was a mistake, we initially used all data, including the 15 initial repetitions. Now we used only 1 repetition per subject. See new Table 4.

Table 2: The unit of measurement for “Measurement Error” and “repeatability” is missing, or? Both outcomes are not well described within the manuscript (and only short in the given reference)

We have included the units in the tables and tried to enhance the statistical analysis section.

Table 4: the denotation “DI/TM” might be changed to “DI or TM”

Done.

Line 276: language: why is a comparison justified? This is not well expressed. Low Spearman’s rho results (especially for the Thomas test) need to be further discussed.

Thank you for the hint. We have rewritten the whole section. 

Line 279 and following: but there are more potential methodological reasons for that.

Thanks for mentioning. We have rephrased the entire discussion, calculated new correlations.

Line 281 and following: Statement not clear

The entire section has been rephrased now.

Line 331: So in other words the comparison is potentially based on invalid data! That should be investigated and clearly stated

The authors are not sure, what comparison you mean, since line 331 is empty. You potentially mean Line 313, which is about the Thomas test. This confusion was based on the wrongly calculated correlations. In the revised manuscript with the newly calculated correlations this does not occur anymore.

Line 334 and following: measurement units?

Included.

Line 356: If these methods are adequate to measure individual training progress, then it would be good to discuss how precise exactly they are. What is the minimal detectable change? Statement regarding the identification of individual training progress might however be beyond the possibilities of this study.

Thank you for the advice. We included data on the measurement precision in the rewritten discussion and were more careful considering the training progress evaluation.

---

## [Decision Letter · Decision Letter 1]

11 Nov 2020

PONE-D-20-15948R1

Intra- and Inter-Rater Reliability of Joint Range of Motion Tests using Tape Measure, Digital Inclinometer and Inertial Motion Capturing

PLOS ONE

Dear Dr. Fräulin,

Thank you for submitting your manuscript to PLOS ONE. After careful consideration, we feel that it has merit but does not fully meet PLOS ONE’s publication criteria as it currently stands. Therefore, we invite you to submit a revised version of the manuscript that addresses the points raised during the review process.

ACADEMIC EDITOR: Thank you for addressing most of the reviewer comments. The quality of your manuscript improved and puts out the strength of your manuscript. Nevertheless, there are still some minor comments left. Please take care of them while revising your manuscript.

We look forward to receiving your revised manuscript.

Kind regards,

Juliane Müller, PhD

Academic Editor

PLOS ONE

Reviewers' comments:

Reviewer's Responses to Questions

**Comments to the Author**

1. If the authors have adequately addressed your comments raised in a previous round of review and you feel that this manuscript is now acceptable for publication, you may indicate that here to bypass the “Comments to the Author” section, enter your conflict of interest statement in the “Confidential to Editor” section, and submit your "Accept" recommendation.

Reviewer #1: (No Response)

2. Is the manuscript technically sound, and do the data support the conclusions?

Reviewer #1: Yes

3. Has the statistical analysis been performed appropriately and rigorously? 

Reviewer #1: Yes

4. Have the authors made all data underlying the findings in their manuscript fully available?

Reviewer #1: Yes

5. Is the manuscript presented in an intelligible fashion and written in standard English?

Reviewer #1: Yes

6. Review Comments to the Author

Reviewer #1: General comments

Thank you for attending to most of my comments on the initial version of the manuscript. The remaining specific comments I’ve provided below are ways in which I still believe the manuscript can be improved, with the line numbers I have provided with respect to the track changes version of your manuscript.

Specific comments

line 121: please remove this p-value regarding the RMSE.

Line 305-306: it might be worthwhile to say in the sentence that the XSens software calculates the relevant joint angles for all parts of the body assessed in the study with the exception of the shoulder test.

Line 360 – 364: I am a little bit unsure what is meant by the comparison of the measurement systems if that is not a validity question. Does this still reflect the revised title and aim of the study and my initial concerns regarding a validity comparison? Or are you using such a comparison to make it clear that these different measuring systems are not directly comparable and shouldn’t be interchanged?

Table 1 – 2: in regards to the measurement error, repeatability and COR scores, you have listed the units as (cm/o). This suggested the reader that you have some new unit of measurement that combines both distance and angles for all of your tests. However, I feel what you’re trying to represent here is that some of these measures are distances and others are joint angles. Therefore, can you make it clearer in these tables which unit of measurement reflects each of the different tests?

7. PLOS authors have the option to publish the peer review history of their article (what does this mean?). If published, this will include your full peer review and any attached files.

Reviewer #1: No

---

## [Author Response · Author response to Decision Letter 1]

11 Nov 2020

Thanks again for you’re your time and effort in reviewing our manuscript. We address the minor revisions in the latest clean version of the manuscript. Therefore, the line numbering is different. I added the correct line numbering in the specific comments.

Reviewer 1:

Specific comments

line 121: please remove this p-value regarding the RMSE.

Done. See line 97.

Line 305-306: it might be worthwhile to say in the sentence that the XSens software calculates the relevant joint angles for all parts of the body assessed in the study with the exception of the shoulder test.

Done. See line 251-254.

Line 360 – 364: I am a little bit unsure what is meant by the comparison of the measurement systems if that is not a validity question. Does this still reflect the revised title and aim of the study and my initial concerns regarding a validity comparison? Or are you using such a comparison to make it clear that these different measuring systems are not directly comparable and shouldn’t be interchanged?

Sorry, that term “comparison” must have been missed in the first revision process. Of course we agree that it is not a validity study. We rather wanted to describe the relationship between the systems by showing the Pearson correlation. In the results section we already changed the wording in the first revision. Now we also changed it in this section, see line 287-289.

Table 1 – 2: in regards to the measurement error, repeatability and COR scores, you have listed the units as (cm/o). This suggested the reader that you have some new unit of measurement that combines both distance and angles for all of your tests. However, I feel what you’re trying to represent here is that some of these measures are distances and others are joint angles. Therefore, can you make it clearer in these tables which unit of measurement reflects each of the different tests?

Thanks for mentioning, we have already described that in the description of the tables, but we agree it should be visible in the table itself. We changed that in the tables.

---

## [Decision Letter · Decision Letter 2]

25 Nov 2020

Intra- and Inter-Rater Reliability of Joint Range of Motion Tests using Tape Measure, Digital Inclinometer and Inertial Motion Capturing

PONE-D-20-15948R2

Dear Dr. Fraeulin,

We’re pleased to inform you that your manuscript has been judged scientifically suitable for publication and will be formally accepted for publication once it meets all outstanding technical requirements.

We are thankful for your patients during this long lasting review process.

Kind regards,

Juliane Müller, PhD

Academic Editor

PLOS ONE

Additional Editor Comments (optional):

Reviewers' comments:

Reviewer's Responses to Questions

**Comments to the Author**

1. If the authors have adequately addressed your comments raised in a previous round of review and you feel that this manuscript is now acceptable for publication, you may indicate that here to bypass the “Comments to the Author” section, enter your conflict of interest statement in the “Confidential to Editor” section, and submit your "Accept" recommendation.

Reviewer #1: All comments have been addressed

2. Is the manuscript technically sound, and do the data support the conclusions?

Reviewer #1: Yes

3. Has the statistical analysis been performed appropriately and rigorously? 

Reviewer #1: Yes

4. Have the authors made all data underlying the findings in their manuscript fully available?

Reviewer #1: Yes

5. Is the manuscript presented in an intelligible fashion and written in standard English?

Reviewer #1: Yes

6. Review Comments to the Author

Reviewer #1: (No Response)

7. PLOS authors have the option to publish the peer review history of their article (what does this mean?). If published, this will include your full peer review and any attached files.

Reviewer #1: **Yes: **Justin Keogh

---

## [Editor Report · Acceptance letter]

27 Nov 2020

PONE-D-20-15948R2 

Intra- and Inter-Rater Reliability of Joint Range of Motion Tests using Tape Measure, Digital Inclinometer and Inertial Motion Capturing 

Dear Dr. Fraeulin:

I'm pleased to inform you that your manuscript has been deemed suitable for publication in PLOS ONE. Congratulations! Your manuscript is now with our production department. 

Kind regards, 

on behalf of

Dr. Juliane Müller 

Academic Editor

PLOS ONE